# Caregivers’ Difficulty in Managing Smartphone Use of Children with Attention-Deficit/Hyperactivity Disorder during the COVID-19 Pandemic: Relationships with Caregiver and Children Factors

**DOI:** 10.3390/ijerph19095194

**Published:** 2022-04-25

**Authors:** Jia-In Lee, Ray C. Hsiao, Ching-Shu Tsai, Cheng-Fang Yen

**Affiliations:** 1Department of Psychiatry, Kaohsiung Medical University Hospital, Kaohsiung 80756, Taiwan; 1050644@kmuh.org.tw; 2Department of Psychiatry, School of Medicine, College of Medicine, Kaohsiung Medical University, Kaohsiung 80708, Taiwan; 3Department of Psychiatry and Behavioral Sciences, University of Washington School of Medicine and Children’s Hospital, Seattle, WA 98105, USA; rhsiao@u.washington.edu; 4Department of Child and Adolescent Psychiatry, Chang Gung Memorial Hospital, Kaohsiung Medical Center, Kaohsiung 83301, Taiwan; 5School of Medicine, Chang Gung University, Taoyuan 33302, Taiwan; 6College of Professional Studies, National Pingtung University of Science and Technology, Pingtung 91201, Taiwan

**Keywords:** attention-deficit/hyperactivity disorder, caregiver, COVID-19, smartphone use, psychological well-being

## Abstract

This study examined the difficulty encountered by caregivers of children with attention-deficit/hyperactivity disorder (ADHD) in managing children’s smartphone use during the coronavirus disease 2019 (COVID-19) pandemic and the caregiver- and children-related factors that influence this difficulty. In total, 252 caregivers of children with ADHD were recruited into this study. The caregivers completed a research questionnaire to provide data regarding the difficulty they encountered in managing the smartphone use of children during the COVID-19 pandemic, their general mental health and parenting styles, and the ADHD and oppositional defiant disorder (ODD) symptoms of the children they are caring for. The results indicated that almost 45% of the caregivers of children with ADHD sometimes or often found it difficult to manage the smartphone use of children with ADHD during the COVID-19 pandemic. For the caregivers, a short duration of education, poor general mental health, unaffectionate/uncaring and overprotective parenting styles, older children, and inattention and ODD symptoms were significantly associated with increased difficulty in managing their children’s smartphone use during the COVID-19 pandemic. On the basis of the relevant factors identified in this study, an intervention should be developed to enhance the skills of caregivers of children with ADHD with respect to the management of children’s smartphone use during the COVID-19 pandemic.

## 1. Introduction

The coronavirus disease 2019 (COVID-19) pandemic has exerted greater adverse effects on the daily lives of children with attention-deficit/hyperactivity disorder (ADHD) than on those without ADHD [1]. Children with ADHD may encounter difficulty in receiving professional evaluations, pharmacological treatments, and psychotherapy during the pandemic [1]. During the lockdown and closure of schools due to the COVID-19 pandemic, emotional dysregulation [2,3], sleep disturbance [1,2], and behavioral problems [4] are prevalent among children with ADHD. Caregivers are responsible for supervising the remote learning of children with ADHD [5] and managing the worsening of their ADHD and emotional and behavioral problems during the period of school closure caused by the pandemic [6]. The multilevel challenges of caring for children with ADHD during the COVID-19 pandemic may affect the mental health of caregivers [6] and caregiver–child interactions [7,8].

During the COVID-19 pandemic, managing the smartphone use of children is a challenge encountered by the caregivers of children with ADHD. Before the COVID-19 pandemic, smartphone overuse was already common among children with ADHD. A study in South Korean children reported that ADHD was the most significant factor related to smartphone overuse [9]. Several biopsychosocial characteristics may contribute to smartphone overuse in children with ADHD, including intolerance to delayed rewards and boredom, a high tendency to give up and habituate repeated positive reinforcement, poor inhibitory control, and difficulty in social adaptation [10,11]. Longitudinal studies have discovered that smartphone overuse predicts low self-esteem [11], loneliness, and depression [12] in children.

The smartphone use of children with ADHD may exacerbate during the COVID-19 pandemic for several reasons. Children are forced to reduce outdoor activities and social interactions for protection against COVID-19. Consequently, children with ADHD may naturally spend more time using their smartphones to relieve boredom and seek fun during the COVID-19 pandemic. Moreover, smartphones can be used to cope with the psychological stress caused by the COVID-19 pandemic [13]. Given that smartphone overuse may exacerbate the ADHD and mood symptoms of children by affecting their sleep hygiene, daily performance, and family interactions [14], the difficulty encountered by caregivers in managing the smartphone use of children with ADHD during the COVID-19 pandemic and the factors related to this problem warrant further investigation.

Under the ecological model [15], both caregiver- and children-related factors can influence the efficacy of caregivers of children with ADHD in managing the smartphone use of children with ADHD. For example, caregivers’ years of education was positively associated with caregivers’ self-efficacy in managing Internet use of children with ADHD [16]. The mental health problems of caregivers may negatively influence their self-efficacy with respect to their communication with children and their management of problematic behaviors in their children [17]. Caregivers’ anxiety was also significantly associated with smartphone/Internet overuse in children with ADHD [18]. The implementation of a democratic parenting style by caregivers may reduce the likelihood of smartphone overuse in children [19], whereas the implementation of an overprotective parenting style by caregivers may increase the likelihood of smartphone overuse in children [20,21]. An affectional and caring parenting style was positively associated with parental self-efficacy in managing Internet use of children with ADHD [16]. Moreover, the ADHD symptoms of children may indicate their levels of self-control and cooperation with caregivers when they are distracted or exhibiting impulsive behavior. The oppositional defiant disorder (ODD) symptoms of children may cause caregivers to experience frustration and low self-confidence when they are trying to manage the electronic media use of children with ADHD [16]. The factors related to the difficulty encountered by the caregivers of children with ADHD in managing the smartphone use of these children during the COVID-19 pandemic warrant further research, and the findings of such research can provide a reference for developing intervention programs.

Through the present study, we examined the caregiver-related factors (demographics, general mental health state, and parenting styles) and children-related factors (demographics, taking medication for ADHD, and ADHD and ODD symptoms) that influence the difficulty encountered by the caregivers of children with ADHD in managing the smartphone use of children during the COVID-19 pandemic. We hypothesized that several factors are associated with an increased difficulty in managing the smartphone use of children for caregivers. Specifically, the caregiver-related factors are a low educational level, poor mental health, an unaffectionate/uncaring parenting style, and an overprotective parenting style; the children-related factors are older age and severe ADHD and ODD symptoms.

## 2. Methods

### 2.1. Participants

The caregivers of children with ADHD, who were aged 6–18 years and were visiting the child psychiatric outpatient clinics of two medical centers in Kaohsiung, Taiwan, and were diagnosed as having ADHD, according to the *Diagnostic and Statistical Manual of Mental Disorders, Fifth Edition* [22], were invited to participate in the present study. Caregivers who had any cognitive impairment (e.g., addictive substance use, schizophrenia, and intellectual disability) that could have prevented them from understanding the purpose and procedure of the present study were excluded. The present study was conducted between August 2021 and January 2022. Between May and July of 2021, Taiwan experienced a severe COVID-19 outbreak, during which schools were closed for the first time since the start of the COVID-19 pandemic to prevent the spread of COVID-19. Children were asked to stay home as much as possible to reduce their risk of contracting COVID-19. After-school classes, cramming schools, sports facilities, and playgrounds for children were also closed. Numerous caregivers were also forced to work at home or to stop working, which increased the time they spent at home. Therefore, these caregivers had more time to interact with their children with ADHD during this period than before the pandemic.

Informed consent was obtained from all participants prior to the assessment conducted in the present study. The participants could seek help from research assistants when they experienced difficulty in completing the questionnaire. In total, 252 caregivers (200 female and 52 male) of children with ADHD participated in this study and completed self-reporting research questionnaires. Their mean age was 42.23 years (SD = 8.34 years); and the mean years of education was 14.18 years (SD = 2.68 years). Regarding the demographic characteristics of children, there were 52 girls and 200 boys with ADHD; their mean age was 10.29 years (SD = 2.94 years). The present study was approved by the Institutional Review Board of Chang Gung Medical Foundation (202002118B0C501) and Kaohsiung Medical University Hospital (KMUHIRB-E(I)-20200408).

### 2.2. Measures

#### 2.2.1. Caregivers’ Difficulty in Managing Smartphone Use of Children with ADHD

One item was used to assess how often the caregivers experienced difficulty in managing the smartphone use of their children with ADHD (“How often did you experience difficulty in managing your child’s smartphone use during school closure and delayed school opening due to the COVID-19 pandemic?”). The caregivers rated the item on a 4-point scale from 0 (never) to 3 (often). The total score indicated the level of difficulty experienced by the caregivers in managing the smartphone use of their children with ADHD during the COVID-19 pandemic.

#### 2.2.2. ADHD and ODD Symptoms

The ADHD and ODD symptoms of children in the month preceding the commencement of the present study were assessed using the traditional Chinese version [23] of the Swanson, Nolan, and Pelham version IV scale (SNAP-IV) [24]. The caregivers rated the items of the subscales of inattention (9 items), hyperactivity/impulsivity (9 items), and ODD symptoms (8 items) on a 4-point Likert scale ranging from 0 (not at all) to 3 (extremely). The Cronbach’s α values of the inattention, hyperactivity/impulsivity, and ODD subscales in the present study were 0.89, 0.90, and 0.92, respectively.

#### 2.2.3. Use of ADHD Medication among Children

The frequency at which the children took medication prescribed for ADHD was assessed using the following question: “How often does your child take medication prescribed for treating ADHD?” The caregivers rated the question on a 4-point scale from 0 (never) to 3 (often). The children who received a score of 3 were classified into the regular ADHD medication use group, whereas those with a score of less than 3 were classified into the non-regular ADHD medication use group.

#### 2.2.4. General Mental Health State of Caregivers

The self-reported general mental health of the caregivers during the week preceding the commencement of the present study was assessed using the 5-item Brief Symptom Rating Scale (BSRS-5) [25]. The BSRS-5 contains five items that assess anxiety, depressive mood, hostility, interpersonal hypersensitivity, and insomnia. The caregivers self-rated the aforementioned five items on a 5-point scale from 0 (not at all) to 4 (extremely). Studies have verified that the BSRS-5 has acceptable psychometric properties [25,26,27]. The Cronbach’s α of the BSRS-5 in the present study was 0.90. The caregivers with a total BSRS-5 score of ≥6 were regarded as having a poor general mental health state [25].

#### 2.2.5. Parenting Behavior

We used the traditional Chinese version [28,29] of the Parental Bonding Instrument (PBI)-Parent Version [30] to measure parenting behaviors in three dimensions, namely parental affection/care (12 items, e.g., “I could make the child feel better when he or she was upset.”), parental overprotection (seven items, e.g., “I tried to control everything that the child did.”), and authoritarian parenting (six items, e.g., “I let the child do the things that he or she liked to do.”). The caregivers rated each item on a 4-point Likert scale ranging from 1 (agree) to 4 (disagree). We reverse-coded the items to facilitate the interpretation process. A high score for the parental affection/care dimension indicated a caregiver was highly affectionate and warm; a high score for the authoritarian parenting dimension indicated that a caregiver highly encouraged behavioral freedom; and a high score for the parental overprotection dimension indicated that a caregiver had a high degree of denial of psychological autonomy. Studies have verified that the traditional Chinese version of PBI has acceptable reliability and validity [29]. The Cronbach’s α values of the parent-reported affection/care, overprotection, and authoritarian parenting dimensions were 0.82, 074, and 0.71, respectively.

#### 2.2.6. Demographics

Data on the sex (0 = female; 1 = male), age, and educational level of the caregivers and the sex (0 = girls; 1 = boys) and age of the children were collected.

### 2.3. Statistical Analysis

Descriptive results are presented as frequencies and numbers (percentages) for categorical variables and as means and standard deviations (SDs) for continuous variables. We examined the skewness and kurtosis of continuous variables and discovered that all of their absolute values were less than 2, indicating normal distributions [31]. A univariate linear regression analysis was conducted to individually examine the associations of caregivers’ demographics, caregivers’ general mental health state, caregivers’ parenting styles, children’s demographics, children’s medication use, and children’s ADHD and ODD symptoms (independent variables) with the difficulty encountered by the caregivers in managing their children’s smartphone use during the COVID-19 pandemic. Because of high collinearity (condition index = 62.676), the associations of caregiver-related and children-related factors with the difficulty encountered by the caregivers in managing their children’s smartphone use during the COVID-19 pandemic were examined using a backward stepwise multivariate linear regression analysis to prevent the statistical problem of collinearity. A two-tailed *p* value of <0.05 indicated statistical significance. Data were analyzed using SPSS version 24.0 (SPSS, Chicago, IL, USA) software.

## 3. Results

Table 1 lists the results pertaining to the difficulty encountered by the caregivers in managing their children’s smartphone use and the caregiver- and children-related factors that influence this difficulty. In total, 91 (36.1%) caregivers reported having poor general mental health; and the mean (SD) PBI scores for the affection/care, overprotection, and authoritarian parenting dimensions were 37.08 (5.16), 13.75 (3.32), and 12.24 (2.67), respectively. Regarding children-related factors, 212 (84.13%) children were on regular medication for treating ADHD; and the children’s mean (SD) scores for the dimensions of inattention, hyperactivity/impulsivity, and ODD symptoms were 12.88 (5.83), 9.93 (6.17), and 9.33 (5.92), respectively.

The proportion of the caregivers who reported that they never, seldom, sometimes, and often encountered difficulty in managing their children’s smartphone use during the COVID-19 pandemic was 29.76%, 25.40%, 23.81%, and 21.03%, respectively. The mean (SD) level of the aforementioned difficulty was 1.36 (1.12).

Table 2 lists the results of the univariate linear regression analysis of the individual associations of caregiver- and children-related factors with the difficulty encountered by the caregivers in managing their children’s smartphone use. The results indicated that poor general mental health (caregiver), an unaffectionate/uncaring parenting style (caregiver), older age (children), and severe inattention and ODD symptoms (children) were significantly associated with a higher level of difficulty encountered by the caregivers in managing their children’s smartphone use during the COVID-19 pandemic.

Table 3 lists the results of the backward multivariate linear regression analysis of the associations of caregiver- and children-related factors with the difficulty encountered by the caregivers in managing their children’s smartphone use. The results indicated that a short duration of education (caregiver), poor general mental health (caregiver), overprotective parenting style (caregiver), older age (children), and severe inattention and ODD symptoms (children) were significantly associated with a higher level of difficulty encountered by the caregivers in managing their children’s smartphone use during the COVID-19 pandemic. The value of the condition index for the present study was 24.609, indicating the absence of collinearity.

## 4. Discussion

In the present study, we discovered that 23.81% and 21.03% of the caregivers of children with ADHD sometimes and often, respectively, encountered difficulty in managing the smartphone use of their children with ADHD during the COVID-19 pandemic. A short duration of education (caregiver), poor general mental health (caregiver), unaffectionate/uncaring and overprotective parenting styles (caregiver), older age (children), and severe inattention and ODD symptoms (children) were significantly associated with a higher level of difficulty encountered by the caregivers in managing the smartphone use of their children during the COVID-19 pandemic.

The present study revealed that almost 45% of caregivers encountered difficulty in managing the smartphone use of their children with ADHD during the COVID-19 pandemic. Because the present study did not assess whether the caregivers experienced a similar difficulty in managing the smartphone use of their children with ADHD before the outbreak of COVID-19, it could not be determined whether the difficulty encountered by the caregivers was already present before the pandemic. However, the difficulty in managing the smartphone use of children with ADHD during the COVID-19 pandemic may occur in conjunction with the overuse of smartphones by children with ADHD. The smartphone use of children with ADHD may exacerbate during the COVID-19 pandemic due to the interactions between individual and environmental factors. Children have increased time for staying home due to the closure of schools, sports grounds, and after-school classes during the pandemic. Consequently, children may naturally spend more time using their smartphones to relieve boredom and seek fun. Compared with children without ADHD, children with ADHD have the tendencies of intolerance to boredom, habituation with repeated positive reinforcement, and poor inhibitory control [10,11], which may exacerbate their smartphone use during their less structured daily lives in the COVID-19 pandemic. Meanwhile, children might experience psychological stress when they faced the drastic changes in their daily lives during the pandemic and when they were worried about the the risk of contracting COVID-19; smartphone use is a common way for children to cope with psychological distress [13]. Alternatively, smartphone/Internet overuse may compromise the health of children with ADHD [1,2,4,32] and increase the difficulty that they may experience in returning to their daily schedules after the mitigation of the pandemic. The difficulty in managing the smartphone use of children may also interfere with the self-efficacy of caregivers and with caregiver–child relationships. The results of the present study indicate that caregivers must receive assistance to help them develop adequate skills for communicating the concept of using smartphones in moderation to children with ADHD.

The present study revealed that the educational level, general mental health state, and unaffectionate/uncaring and overprotective parenting styles of the caregivers were significantly associated with the difficulty encountered by the caregivers in managing the smartphone use of children with ADHD during the COVID-19 pandemic. A lower educational level may indicate less knowledge of electronic media and less capacity for monitoring and managing the smartphone/Internet use of children [16], thus increasing the difficulty of caregivers in communicating the concept of managing smartphone use to children with ADHD. Poor general mental health was significantly associated with the difficulty encountered by the caregivers of children with ADHD in managing the smartphone use of their children. The association was partially accounted for by the negative influence of mental health problems on caregivers’ self-efficacy with respect to their communication with children and their management of problematic behaviors in their children [17]. Studies have revealed that a high proportion of caregivers of children with ADHD experience mental health problems [6,33,34,35,36]. The difficulty in managing the behaviors of children with ADHD (e.g., smartphone use) may further affect the mental health of caregivers during the COVID-19 pandemic. Alternatively, mental health problems may also affect the cognitive function, communication skills, and anger management of caregivers, which are essential for the successful management of the problematic behaviors of children with ADHD.

The present study revealed that unaffectionate/uncaring and overprotective parenting styles were significantly associated with a higher level of difficulty in managing the smartphone use of children during the COVID-19 pandemic among the caregivers of children with ADHD. Parenting styles reflect the attitudes and behaviors of caregivers toward their children during the early stages of their children’s development [37]. Overprotective parenting reflects the denial of the psychological autonomy of children by their caregivers [38]; children with low psychological autonomy may lose their self-control over their smartphone use during a period characterized by the absence of external controls associated with school regulations and other routine activities. Research on the general children population has confirmed that an overprotective parenting style increases the likelihood of smartphone overuse in children [20,21]. Research also found that an affectional and caring parenting style was positively associated with parental self-efficacy in managing Internet use of children with ADHD [16] and low caregivers’ affiliate stigma [39]. Caregivers who are uncaring/unaffectionate toward children with ADHD may experience increased difficulty in engaging in caregiver–child communication; consequently, caregivers and their children with ADHD may lack a favorable foundation for collectively developing a consensus on smartphone use during the COVID-19 pandemic.

The present study revealed that among the children-related factors, older age and severe inattention and ODD symptoms were significantly associated with caregivers encountering increased difficulty in managing the smartphone use of their children during the COVID-19 pandemic. Older children are more likely to have their own smartphone and have social communication needs involving their peers; thus, they have an increased risk of smartphone overuse relative to younger children [40]. Older children may also adopt a firm attitude to argue with their caregivers over their right to use smartphones. These developmental demands and changes may increase the difficulty encountered by caregivers in managing the smartphone use of children during the pandemic. Children with severe inattention symptoms may find more pleasure in using smartphones than in participating in other sedentary activities (e.g., reading a book); thus, they are susceptible to smartphone overuse when they are asked to stay indoors during the pandemic. For children with ADHD, inattention symptoms may also make it more difficult for them to remember information and cooperate with caregivers on the management of smartphone use. A high proportion of children with ADHD have comorbid ODD symptoms, which may compromise the prognosis of ADHD [41]. Children with ADHD who have ODD symptoms may argue with their caregivers and refuse to comply with their caregivers’ requests. ODD symptoms of children may also exacerbate caregivers’ frustration and low self-confidence in managing the electronic media use of children with ADHD [16]. Thus, ODD symptoms inevitably increase the difficulty encountered by caregivers in getting their children with ADHD to follow rules regarding smartphone use during the pandemic.

The findings of the present study highlight the value of developing strategies for the difficulty in managing children’s smartphone use among caregivers of children with ADHD during the COVID-19 pandemic. The first is to enhance caregivers’ skills for managing the smartphone use of children with ADHD. A meta-analysis has confirmed that psychological interventions based on a cognitive behavioral model for Internet/smartphone addiction among adolescents may help to reduce addiction severity [42]. Professionals can use the standard instrument measuring and help increase caregivers’ self-efficacy in managing smartphone use of children with ADHD [43]. Moreover, given that caregivers’ parenting styles and mental health related to the difficulty in managing children’s smartphone use, assisting caregivers in modifying their parental styles and maintaining mental health may help reduce the difficulty that they encounter in managing the smartphone use of children. Caregivers’ mental health should be routinely assessed to detect mental health problems early. Timeliness in providing psychological and pharmacological assistance may contribute to the improvement of mental health in caregivers. However, how to conduct the psychological interventions for caregivers to enhance their skills for managing children’s smartphone use, modify parental styles, and maintain mental health is a great challenge during the COVID-19 pandemic. Under the circumstances in the pandemic, when face-to-face interventions are not possible, modified models of health service provision should be developed. For example, intervention programs using telepsychiatry or telepsychology may provide feasible and convenient behavioral interventions for caregivers of children with ADHD [44]. Moreover, the creation of a structured daily schedule and reinforcement system and the implementation of child-appropriate activities are the basic steps that caregivers can take to assist children with ADHD in reducing their smartphone use. Regarding children’s ADHD symptoms, regular online consultations and online therapy sessions must also be implemented to prevent the worsening of ADHD symptoms and the corresponding occurrence of smartphone overuse. Given that pharmacological treatment significantly reduces the risk of negative outcomes in individuals with ADHD [45], continuing effective pharmacological treatment for ADHD during the COVID-19 pandemic is recommended [1]. Low caregiver education and children’s age were significantly associated with caregivers’ difficulty in managing children’s smartphone use. Although caregivers’ education level and children’s age are not modifiable, intervention programs for enhancing caregivers’ skills for managing children’s smartphone use should take these factors into consideration. For example, the programs should modify the wording and the way of explaining to help less educated caregivers learn the skills. Intervention programs should be also specified for caregivers according to the developmental stage of children.

This study has several limitations. First, the participants were enrolled from pediatric psychiatric outpatient clinics. Thus, the results of this study cannot be generalized to caregivers who do not seek medical assistance for treating their children with ADHD. Second, the cross-sectional design of the present study limited our ability to draw conclusions regarding the temporal relationships between the difficulty encountered in managing the smartphone use of children and the poor mental health and parenting styles of caregivers. Third, we obtained all of our data from only caregivers, which could have resulted in the problem of shared-method variance due to the use of a single information source. Especially, children may have their own perception of parenting styles that may be not totally the same as the caregivers’ perceptions. The PBI has a child-report version [28,30] and can be used to compare the difference in the association between caregivers’ difficulty in managing children’s smartphone use and the parenting styles from both caregiver and childrenreports. Fourth, the present study did not recruit the caregivers of children without ADHD for comparative purposes and thus could not ascertain whether caregivers of children without ADHD experience similar difficulty in managing children’s smartphone use. This study neither assessed whether the caregivers experienced a similar difficulty in managing the smartphone use of their children with ADHD before the outbreak of COVID-19. The study could not be determined whether the difficulty encountered by the caregivers was already present before the pandemic. Future studies should examine whether the factors related to the difficulty encountered in managing the smartphone use of children with ADHD (as identified in the present study) could also apply to all caregivers of children after the remission of the COVID-19 pandemic.

## 5. Conclusions

The findings of this study revealed that a high proportion of caregivers of children with ADHD encountered difficulty in managing the smartphone use of their children during the COVID-19 pandemic. Enhancing caregivers’ skills for managing the smartphone use of children with ADHD is of utmost importance. How to deliver training programs for caregivers of children with ADHD during the pandemic warrants a well-planned system based on evidence. Mental health professionals should consider the caregiver and child factors related to the difficulty of managing the smartphone use of children (as identified in the present study). Intervention programs using telepsychiatry or telepsychology may provide services for caregivers who have the need for mental health services and for children who need continuing psychotherapeutic and psychopharmacological intervention for ADHD. Professionals should also consider caregivers’ education level and children’s age and specify intervention programs for individual needs. Professionals should also help caregivers reflect on their parenting styles and develop the parenting styles that help increase the ability to manage smartphone use of children with ADHD. 

## Figures and Tables

**Table 1 ijerph-19-05194-t001:** Caregivers’ Difficulty in Managing Smartphone Use of Children and Caregiver- and Children-Related Factors (*n* = 252).

	*n* (%)	Mean (SD)	Range
Caregivers’ difficulty in managing child’s smartphone use			
Never	75 (29.76)		
Seldom	64 (25.40)		
Sometimes	60 (23.81)		
Often	53 (21.03)		
Level of difficulty		1.36 (1.12)	0–3
*Caregivers*			
Gender			
Female	200 (79.37)		
Male	52 (20.63)		
Age (years)		42.23 (8.34)	23–77
Caregivers’ educational level (years)		14.18 (2.68)	6–20
General mental health state			
Good	161 (63.89)		
Poor	91 (36.11)		
Parenting style			
Affectionate/caring		37.08 (5.16)	19–47
Overprotective		13.75 (3.32)	7–22
Authoritarian parenting		12.24 (2.67)	6–21
*Children*			
Girls	52 (20.63)		
Boys	200 (79.37)		
Age (years)		10.29 (2.94)	6–23
Taking of medication for ADHD			
No or irregular	40 (15.87)		
Regular	212 (84.13)		
Inattention symptoms		12.88 (5.83)	0–27
Hyperactivity/impulsivity symptoms		9.93 (6.17)	0–27
Oppositional ODD symptoms		9.33 (5.92)	0–23

ADHD, attention-deficit/hyperactivity disorder; ODD, oppositional defiant disorder.

**Table 2 ijerph-19-05194-t002:** Univariate Linear Regression Analysis of Associations of Caregiver- and Children-Related Factors with Caregivers’ Difficulty in Managing Smartphone Use of Children.

	B (SE)
Caregivers’ gender	−0.164 (0.174)
Caregivers’ age	0.007 (0.008)
Caregivers’ education	−0.032 (0.026)
Caregivers’ general mental health state	0.467 (0.144) **
Caregivers’ affectionate/caring parenting style	−0.046 (0.013) **
Caregivers’ overprotective parenting style	0.035 (0.021)
Caregivers’ authoritarian parenting style	0.020 (0.026)
Children’s gender	−0.175 (0.174)
Children’s age	0.099 (0.029) **
Children’s taking medication for ADHD	0.310 (0.192)
Children’s inattention symptoms	0.046 (0.012) ***
Children’s hyperactivity/impulsivity symptoms	0.021 (0.011)
Children’s ODD symptoms	0.051 (0.012) ***

ADHD, attention-deficit/hyperactivity disorder; ODD, oppositional defiant disorder. ** *p* < 0.01; *** *p* < 0.001.

**Table 3 ijerph-19-05194-t003:** Associations of Children’s Age, Inattention and ODD Symptoms with Caregivers’ Difficulty to Manage Child’s Smartphone Use: Backward Stepwise Multivariate Linear Regression Analysis.

	B (SE)
Caregivers’ years of education	−0.048 (0.025) *
Caregivers’ general mental health state	0.294 (0.148) *
Caregivers’ overprotective parenting	0.041 (0.020) *
Children’s age	0.119 (0.027) ***
Children’s inattention symptoms	0.029 (0.014) *
Children’s ODD symptoms	0.028 (0.014) *

ADHD: attention-deficit/hyperactivity disorder; ODD: oppositional defiant disorder. * *p* < 0.05; *** *p* < 0.001.

## Data Availability

The data used in this study are available upon reasonable request to the corresponding authors.

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
