# Peer review of "Caregivers’ Difficulty in Managing Smartphone Use of Children with Attention-Deficit/Hyperactivity Disorder during the COVID-19 Pandemic: Relationships with Caregiver and Children Factors"

_ijerph, 2022, doi:10.3390/ijerph19095194_

Round 1
Reviewer 1 Report
Dear authors,
The presented article deals with a very interesting and valuable topic, which is mental health related to the use of electronic devices in children with ADHD.
- The Introduction provides an adequate overview of the current situation regarding ADHD and COVID-19. There are sufficient previous studies to support and substantiate the wording.
- The Methods section is very comprehensive and the instruments used and the steps followed in the assessment are well understood.
However, the section "2.1 Participants" is incomplete. The participant information on lines 197-206 should be in section 2.1, not in section "3. Results".
- The Results section is presented in an orderly and clear manner. The results are detailed in a clear and precise manner.
- The discussion, although it is a good summary of the results of the research, does not relate to the previous studies set out in the introduction.
The discussion should contain a reflection of the results and the previous studies analysed.
- The conclusions are very brief. Given the interest of the study and the encouraging results obtained, such a short conclusion section is not understandable.
The references are appropriate to the format of the journal. No corrections are necessary.
After these suggestions for improvement, the article could be published, but they need to be carefully considered. All suggested improvements will add value to the article.

Author Response
We appreciated your valuable comments. As discussed below, we have revised our manuscript with underlines based on your suggestions. Please let us know if we need to provide anything else regarding this revision.
Comment 1
The presented article deals with a very interesting and valuable topic, which is mental health related to the use of electronic devices in children with ADHD.
Response
Thank you for your positive comment.
Comment 2
The Introduction provides an adequate overview of the current situation regarding ADHD and COVID-19. There are sufficient previous studies to support and substantiate the wording.
Response
Thank you for your positive comment.
Comment 3
The Methods section is very comprehensive and the instruments used and the steps followed in the assessment are well understood. However, the section "2.1 Participants" is incomplete. The participant information on lines 197-206 should be in section 2.1, not in section "3. Results".
Response
We moved the demographic characterization of caregivers and children from “results” section into the “method – participants section.” Please refer to line 123-128.
Comment 4
The Results section is presented in an orderly and clear manner. The results are detailed in a clear and precise manner.
Response
Thank you for your positive comment.
Comment 5
The discussion, although it is a good summary of the results of the research, does not relate to the previous studies set out in the introduction. The discussion should contain a reflection of the results and the previous studies analyzed.
Response
Thank you for your comment. We revised the discussion section and reflected the results of previous studies as below.
“The smartphone use of children with ADHD may exacerbate during the COVID-19 pandemic due to the interactions between individual and environmental factors. Children have increased time for staying home due to the closure of schools, sports ground, and after-school classes during the pandemic. Consequently, children may naturally spend more time using their smartphones to relieve boredom and seek fun. Compared with children without ADHD, children with ADHD have the tendencies of intolerance to boredom, habituation with repeated positive reinforcement, and poor inhibitory control [10,11], which may exacerbate their smartphone use during the less structured daily lives in the COVID-19 pandemic. Meanwhile, children might experience psychological stress when they faced the drastic changes in daily lives during the pandemic and worried the risk of contracting COVID-19; smartphone use is a common way for children to cope with psychological distress [13]. Alternatively, smartphone/internet overuse may compromise the health of children with ADHD [1,2,4,32] and increase the difficulty that they may experience in returning to their daily schedules after the mitigation of the pandemic. The difficulty in managing the smartphone use of children may also interfere with the self-efficacy of caregivers and with caregiver–child relationships. The results of the present study indicate that caregivers must receive assistance to help them develop adequate skills for communicating the concept of using smartphones in moderation to children with ADHD.” Please refer to line 257-275.
“A lower educational level may indicate less knowledge of electronic media and less capacity for monitoring and managing the smartphone/internet use of children [16],” Please refer to line 279-281.
“The association was partially accounted for the negative influence of mental health problems on caregivers’ self-efficacy with respect to their communication with children and their management of problematic behaviors in their children [17].” Please refer to line 285-287.
“Research on the general children population has confirmed that an overprotective parenting style increases the likelihood of smartphone overuse in children [17,18]. Research also found that an affectional and caring parenting style was positively associated with parental self-efficacy in managing Internet use of children with ADHD [19] and low caregivers’ affiliate stigma [39].” Please refer to line 303-307.
“ODD symptoms of children may also exacerbate caregivers’ frustration and low self-confidence in managing the electronic media use of children with ADHD [16].” Please refer to line 329-330.
Comment 6
The conclusions are very brief. Given the interest of the study and the encouraging results obtained, such a short conclusion section is not understandable.
Response
Thank you for your comment. We rewrote the Conclusion section as below. Please refer to line 392-405.
“The findings of this study revealed that a high proportion of caregivers of children with ADHD encountered difficulty in managing the smartphone use of their children during the COVID-19 pandemic. Enhancing caregivers’ skills for managing the smartphone use of children with ADHD is of uttermost importance. How to deliver training programs for caregivers of children with ADHD during the pandemic warrants a well-planned system based on evidence. Mental health professionals should consider the caregiver and child factors related to the difficulty of managing the smartphone use of children (as identified in the present study). Intervention programs using telepsychiatry or telepsychology may provide services for caregivers who have the need for mental health services and children who need continuing psychotherapeutic and psychopharmacological intervention for ADHD. Professional should also consider caregivers’ education level and children’s age and specify intervention programs for individual needs. Professionals should also help caregivers reflect their parenting styles and develop the parenting styles that help increase the ability to manage smartphone use of children with ADHD.”
Comment 7
The references are appropriate to the format of the journal. No corrections are necessary.
Response
Thank you for your positive comment.
Comment 8
After these suggestions for improvement, the article could be published, but they need to be carefully considered. All suggested improvements will add value to the article.
Response
Thank you for your positive comment.
Reviewer 2 Report
Observational study that examined factors associated with problematic negotiation of smartphone use between caregivers and young people with ADHD during the height of the COVID pandemic. Participants were a convenience sample of attenders at one of two child psychiatric outpatient clinics in Taiwan. The primary outcome measure was a caregiver self-reported single item scale rating difficulties in managing smartphone use. Other measures were caregiver report of severity of ADHD and ODD symptoms, child’s adherence to ADHD medication, caregiver wellbeing (Brief Symptom Rating Scale) and parenting style (Parental Bonding Instrument-parent version). Four out of five respondents were mothers. Nearly half the sample reported at least some level of difficulty in managing smartphone use. The authors reported two types of regression analysis of the data. In the first, problems in managing smartphone use were associated with lower caregiver wellbeing, older child age, and higher levels of child inattention and ODD symptoms. Affectionate caring parenting style was protective. In the second regression analysis problems in managing smartphone use were associated with lower caregiver education, poorer caregiver wellbeing, overprotective parenting style, older child, higher levels of child inattention and ODD symptoms. The authors concluded that during the pandemic problems in managing smartphone use in this population were common. Clinicians should address the associated factors identified in the study to assist caregivers and enhance their skills in managing smartphone use.
Specific comments:
- The study is topical as the issue has attracted media attention
- The study design does not permit inferences about how findings relate to ADHD or the COVID pandemic, as there were comparison data for neither.
- Owing to #2, the title of the paper is misleading.
- What does ‘and related factors’ refer to in the title?
- The study hypotheses curiously align exactly with the study findings. This does raise some doubt whether the study hypotheses were formulated prospectively.
- As all measures were completed by the caregiver, there is a substantial risk of informant bias.
- It is not clear to me how the factors associated with problems in managing smartphone use might be specifically addressed in an intervention. Some, such as level of caregiver education and the child’s age, are not modifiable. Some are related to the effective management of the ADHD and ODD, which is presumably the core business of the clinic. This leaves caregiver wellbeing and parenting style. How may they be targeted in an outpatient setting?
Author Response
We appreciated your valuable comments. As discussed below, we have revised our manuscript with underlines based on your suggestions. Please let us know if we need to provide anything else regarding this revision.
Comment 1
The study is topical as the issue has attracted media attention
Response
Smartphone overuse in children and the difficulty faced by caregivers in managing children’s smartphone use are issues that receive media attention. This is principally because children’s smartphone overuse would negatively impact health of children. We expect that the results of this study could add knowledge to the field of prevention and intervention for smartphone overuse in children with ADHD.
Comment 2
The study design does not permit inferences about how findings relate to ADHD or the COVID pandemic, as there were comparison data for neither.
Response
In this study, we recruited caregivers of children with the clinical diagnosis of ADHD. We also assessed their difficulty encountered by in managing their child’ smartphone use during the COVID-19 pandemic and examined its associations with caregiver and child factors, including children’s ADHD symptoms. Therefore, we believed that this study focused on caregivers of children with ADHD during the COVID-19 pandemic. However, we agree that further study is needed to examine whether the results of this study can be generalized to caregivers of children without ADHD and caregivers of children with ADHD after the remission of the pandemic. We added it into the limitations of this study as below. Please refer to line 380-390.
“Fourth, the present study did not recruit the caregivers of children without ADHD for comparative purposes and thus could not ascertain whether caregivers of children without experience the similar difficulty in managing children’s smartphone use. This study neither assessed whether the caregivers experienced a similar difficulty in managing the smartphone use of their children with ADHD before the outbreak of COVID-19. The study could not be determined whether the difficulty encountered by the caregivers was already present before the pandemic. Future studies should examine whether the factors related to the difficulty encountered in managing the smartphone use of children with ADHD (as identified in the present study) could also apply to all caregivers of children after the remission of the COVID-19 pandemic.”
Comment 3
Owing to #2, the title of the paper is misleading.
Response
The title did focus on the aims of this study and study sample (caregivers’ difficulty in managing their child’s smartphone use during the COVID-19 pandemic). Therefore, there should be no misleading in the title.
Comment 4
What does ‘and related factors’ refer to in the title?
Response
To make the meaning of the title clearer, we revised the title into “Caregivers’ Difficulty in Managing Smartphone Use of Children with Attention-Deficit/Hyperactivity Disorder During the COVID-19 Pandemic: Relationships with Caregiver and Children Factors.” Please refer to line 1-4.
Comment 5
The study hypotheses curiously align exactly with the study findings. This does raise some doubt whether the study hypotheses were formulated prospectively.
Response
We made the hypotheses based on the results of previous studies described in Introduction of this manuscript.
- Caregivers’ years of education was positively associated with caregivers’ self-efficacy in managing internet use of children with ADHD [16]. Please refer to line 75-77.
- The mental health problems of caregivers may negatively influence their self-efficacy with respect to their communication with children and their management of problematic behaviors in their children [17]. Please refer to line 77-79.
- The implementation of a democratic parenting style by caregivers may reduce the likelihood of smartphone overuse in children [19], whereas the implementation of an overprotective parenting style by caregivers may increase the likelihood of smartphone overuse in children [20,21]. Affectional and caring parenting style was positively associated with parental self-efficacy in managing Internet use of children with ADHD [16]. Please refer to line 80-85.
- ADHD symptoms of children may indicate their levels of self-control and cooperation with caregivers when they are distracted or exhibiting impulsive behavior. The oppositional defiant disorder (ODD) symptoms of children may cause caregivers to experience frustration and low self-confidence when they are trying to manage the electronic media use of children with ADHD [16]. Please refer to line 85-89.
Comment 6
As all measures were completed by the caregiver, there is a substantial risk of informant bias.
Response
- We agreed that the data from a single information source might result in the problem of shared-method variance. It has been listed in the original version of the manuscript (also in the revised manuscript, line 375-376).
- In addition to the original contents, we added new sentences discussing about the possible role of parenting styles from various sources of reports as below. Please refer to line 376-380.
“Especially, children may have their perception of parenting styles that may be not totally the same as caregivers’ perception. The PBI has a child-report version [28,30] and can be used to compare the difference in the association between caregivers’ difficulty in managing children’s smartphone and parenting styles from caregiver- and children-report.”
Comment 7
It is not clear to me how the factors associated with problems in managing smartphone use might be specifically addressed in an intervention. Some, such as level of caregiver education and the child’s age, are not modifiable. Some are related to the effective management of the ADHD and ODD, which is presumably the core business of the clinic. This leaves caregiver wellbeing and parenting style. How may they be targeted in an outpatient setting?
Response
Thank you for your comment. We revised the contents of the implications as below. Please refer to line 333-368.
“The findings of the present study highlight the value of developing strategies for the difficulty in managing children’s smartphone use among caregivers of children with ADHD during the COVID-19 pandemic. The first is to enhance caregivers’ skills for managing the smartphone use of children with ADHD. A meta-analysis has confirmed that psychological interventions based on cognitive behavioral model for Internet/smartphone addiction among adolescents may help to reduce addiction severity [42]. Professionals can use the standard instrument measuring and help increase caregivers’ self-efficacy in managing smartphone use of children with ADHD [43]. Moreover, given that caregivers’ parenting styles and mental health related to the difficulty in managing children’s smartphone use, assisting caregivers in modifying their parental styles and maintaining mental health may help reduce the difficulty that they encounter in managing the smartphone use of children. Caregivers’ mental health should be routinely assessed to early detect mental health problems. Timely providing psychological and pharmacological assistance may contribute to the improvement of mental health in caregivers. However, how to conduct the psychological interventions for caregivers to enhance their skills for managing children’s smartphone use, modify parental styles, and maintain mental health is a great challenge during the COVID-19 pandemic. Under the circumstances in the pandemic, when face-to-face interventions are not possible, modified models of health service provision should be developed. For example, intervention programs using telepsychiatry or telepsychology may provide feasible and convenient behavioral interventions for caregivers of children with ADHD [44]. Moreover, the creation of a structured daily schedule and reinforcement system and the implementation of child-appropriate activities are the basic steps that caregivers can take to assist children with ADHD in reducing their smartphone use. Regarding children’s ADHD symptoms, regular online consultations and online therapy sessions must also be implemented to prevent the worsening of ADHD symptoms and the corresponding occurrence of smartphone overuse. Given that pharmacological treatment significantly reduces the risk of negative outcomes in individuals with ADHD [45], continuing effective pharmacological treatment for ADHD during the COVID-19 pandemic is recommended [1]. Low caregiver education and children’s age were significantly associated with caregivers’ difficulty in managing children’s smartphone use. Although caregivers’ education level and children’s age are not modifiable, intervention programs for enhancing caregivers’ skills for managing children’s smartphone use should take these factors into consideration. For example, the programs should modify the wording and the way of explaining to help less lectured caregivers learn the skills. Intervention programs should be also specified for caregivers according to the developmental stage of children.”
Reviewer 3 Report
Demographic characterization of caregivers should be in the “method – participants section” and not in “results” section as it is.
In “discussion” section it would be pertinent and important to describe how to assist or help caregivers to develop adequate skills to deal with their children, with ADHD, overuse of smartphones and perhaps to relate that with other educational parenting aspects.
The article is pertinent, actual, scientifically grounded, and interesting.

Author Response
We appreciated your valuable comments. As discussed below, we have revised our manuscript with underlines based on your suggestions. Please let us know if we need to provide anything else regarding this revision.
Comment 1
Demographic characterization of caregivers should be in the “method – participants section” and not in “results” section as it is.
Response
Thank you for your comment. We moved the demographic characterization of caregivers and children from “results” section into the “method – participants section.” Please refer to line 123-128.
Comment 2
In “discussion” section it would be pertinent and important to describe how to assist or help caregivers to develop adequate skills to deal with their children, with ADHD, overuse of smartphones and perhaps to relate that with other educational parenting aspects.
Response
Thank you for your comment. We revised the contents of the implications as below. Please refer to line 333-368.
“The findings of the present study highlight the value of developing strategies for the difficulty in managing children’s smartphone use among caregivers of children with ADHD during the COVID-19 pandemic. The first is to enhance caregivers’ skills for managing the smartphone use of children with ADHD. A meta-analysis has confirmed that psychological interventions based on cognitive behavioral model for Internet/smartphone addiction among adolescents may help to reduce addiction severity [42]. Professionals can use the standard instrument measuring and help increase caregivers’ self-efficacy in managing smartphone use of children with ADHD [43]. Moreover, given that caregivers’ parenting styles and mental health related to the difficulty in managing children’s smartphone use, assisting caregivers in modifying their parental styles and maintaining mental health may help reduce the difficulty that they encounter in managing the smartphone use of children. Caregivers’ mental health should be routinely assessed to early detect mental health problems. Timely providing psychological and pharmacological assistance may contribute to the improvement of mental health in caregivers. However, how to conduct the psychological interventions for caregivers to enhance their skills for managing children’s smartphone use, modify parental styles, and maintain mental health is a great challenge during the COVID-19 pandemic. Under the circumstances in the pandemic, when face-to-face interventions are not possible, modified models of health service provision should be developed. For example, intervention programs using telepsychiatry or telepsychology may provide feasible and convenient behavioral interventions for caregivers of children with ADHD [44]. Moreover, the creation of a structured daily schedule and reinforcement system and the implementation of child-appropriate activities are the basic steps that caregivers can take to assist children with ADHD in reducing their smartphone use. Regarding children’s ADHD symptoms, regular online consultations and online therapy sessions must also be implemented to prevent the worsening of ADHD symptoms and the corresponding occurrence of smartphone overuse. Given that pharmacological treatment significantly reduces the risk of negative outcomes in individuals with ADHD [45], continuing effective pharmacological treatment for ADHD during the COVID-19 pandemic is recommended [1]. Low caregiver education and children’s age were significantly associated with caregivers’ difficulty in managing children’s smartphone use. Although caregivers’ education level and children’s age are not modifiable, intervention programs for enhancing caregivers’ skills for managing children’s smartphone use should take these factors into consideration. For example, the programs should modify the wording and the way of explaining to help less lectured caregivers learn the skills. Intervention programs should be also specified for caregivers according to the developmental stage of children.”
Comment 3
The article is pertinent, actual, scientifically grounded, and interesting.
Response
Thank you for your positive comment.